# A live cell biosensor protocol for high-resolution screening of therapy-resistant cancer cells

**Viral D. Oza** [1,2], **Colin S. Williams** [1,2], **Jessica S. Blackburn** [1,2*]

**1** Department of Molecular and Cellular Biochemistry, University of Kentucky, Lexington, Kentucky, United States of America, **2** Markey Cancer Center, University of Kentucky, Lexington, Kentucky, United States of America

* jsblackburn@uky.edu

## Abstract

The Genetically Encoded Death Indicator (GEDI) is a ratiometric, dual-fluorescence biosensor that enables real-time detection of cell death through calcium influx. Originally developed for use in neurodegeneration models, GEDI can be applied to cancer cells to quantify therapy-induced death at single-cell resolution. This protocol details how to generate GEDI-expressing cancer cell lines, empirically determine stress-induced GEDI thresholds using radiation or chemotherapeutic agents, and perform time-resolved imaging and image analysis to track cell fate. This workflow is optimized for high-throughput drug and radiation screening in heterogeneous populations and is especially useful for identifying chemo- and radio-resistant subclones. Key limitations include the need for empirical GEDI threshold calibration for each treatment condition and careful standardization of imaging parameters. The protocol outputs include GEDI ratio values, single-cell time-of-death annotations, and whole-cell morphological data in parallel, which can be linked to downstream applications such as FACS-based isolation of live or dying subpopulations, transcriptomic profiling of resistant clones, or in vivo validation using xenografts or organotypic slice culture.

## Introduction

Evasion of apoptosis is a hallmark of cancer and measuring apoptotic events is a key determinant of treatment response [1–3]. Programmed cell death can occur as early as 30 minutes after therapy exposure and is influenced by multiple intrinsic and extrinsic pathways, making precise measurement of cell death essential for identifying new vulnerabilities in cancer cells [4,5]. Accurately capturing these events is especially important in drug and radiation screening, where the ability to quantify therapy-induced death is critical for both evaluating treatment efficacy and for detecting resistant subpopulations within heterogeneous tumors.

The heterogeneity within cancer cell lines is at odds with current high-throughput screening (HTS) methods, which typically rely on bulk population readouts of

**Data availability statement:** Data Availability Statement: All data underlying the findings are included within the manuscript and its Supporting Information files. Representative raw and post-processed imaging datasets, along with all analysis code, are publicly available in the GitHub repository "GEDI_Cancer". Full raw imaging datasets are available from the corresponding author without restriction. Associated content io.protocol DOI: https://dx. doi.org/10.17504/protocols.io.eq2ly4d7qlx9/v1.

**Funding:** National Institutes of Health, Grant/ Award Numbers R37CA227656 (to J.S.B.) and F99CA294265 (to V.D.O.) [https://www. nih.gov/] and the Kentucky Pediatric Cancer Research Trust Fund PON27282400002665 [https://www.chfs.ky.gov/agencies/dph/dpqi/ cdpb/Pages/pcrtf.aspx]. This research is also supported by the Flow Cytometry and Immune Monitoring Shared Resource of the Markey Cancer Center (P30CA177558) [https:// ukhealthcare.uky.edu/markey-cancer-center/ research/srf/flow]. The funders had no role in study design, data collection and analysis, decision to publish, or preparation of the manuscript.

**Competing interests:** The authors have declared that no competing interests exist.

cell death [6]. Within a single population, individual cells may vary widely in their responses, with some succumbing quickly to therapy while others survive due to intrinsic mutations or adaptive stress responses [7,8]. Such heterogeneity has emerged as a major hurdle in cancer treatment, as resistant subclones can evade therapy and ultimately drive relapse. Bulk assays that average outcomes across populations often mask these minority populations, obscuring important information about treatment failure. A method to track cell death with single-cell resolution and precise timing is needed to fully capture how different cells in a heterogeneous tumor respond to therapy. However, conventional cell viability and cytotoxicity assays have significant limitations in this regard. Most HTS assays use static timepoints (24–72 hours) that miss early or transient events. Traditional luminescence-based assays like ATP or caspase activity reporters require cell lysis, providing only a snapshot of cell death and losing spatial or temporal context. Annexin V detection can distinguish apoptotic and dead cells, but requires dyes, flow cytometry, or microscopy at fixed time points, making it less compatible for continuous monitoring. Collectively, these traditional methods lack the ability to monitor cell death dynamically in live, individual cells, making it challenging to resolve mixed responses or rare populations. To overcome these limitations, new tools are needed that combine the throughput of population-based assays with the resolution of live-cell, single-cell imaging.

The Genetically Encoded Death Indicator (GEDI) directly addresses these challenges. GEDI is a dual-fluorescence ratiometric reporter initially developed in neuro-degeneration models to measure cell death based on calcium influx [9]. In its original form, GEDI links a constitutive fluorescent reporter (mApple) with a calcium sensor (GC150) through a self-cleaving peptide (P2A) (Fig 1A). GC150 is activated upon an unregulated $Ca^{2+}$ influx, indicating a disruption in membrane integrity (Fig 1B), while mApple provides a whole-cell morphological reference. Normalizing the fluorescence intensity of GC150 to mApple provides a "GEDI ratio", which serves as an absolute per-cell measure of $Ca^{2+}$ dysregulation that provides a quantitative marker of cell death at single-cell resolution (Fig 1C).

Here, we present a protocol that uses the GEDI biosensor to track therapy-induced death in heterogeneous cancer cell populations. The workflow describes how to generate stable GEDI-expressing lines, empirically determine the GEDI threshold for radiation or drug-induced cell death and perform time-resolved imaging of GEDI-expressing cells with automated cell tracking. By capturing longitudinal phenotypic data in conjunction with real-time cell death events, this approach enables deeper insights into dynamic resistance, clonal evolution, and bystander effects. The protocol is designed for high-throughput drug and radiation screens and supports downstream applications where additional investigations on resistant populations are required, such as *in vivo* or *ex vivo* validation, next-generation sequencing analyses, or cell-specific marker identification.

## Materials and methods

The protocol described in this peer-reviewed article is published on protocols.io(dx. doi.org/10.17504/protocols.io.eq2ly4d7qlx9/v1) and is included for printing purposes as **S1 File** with this article.

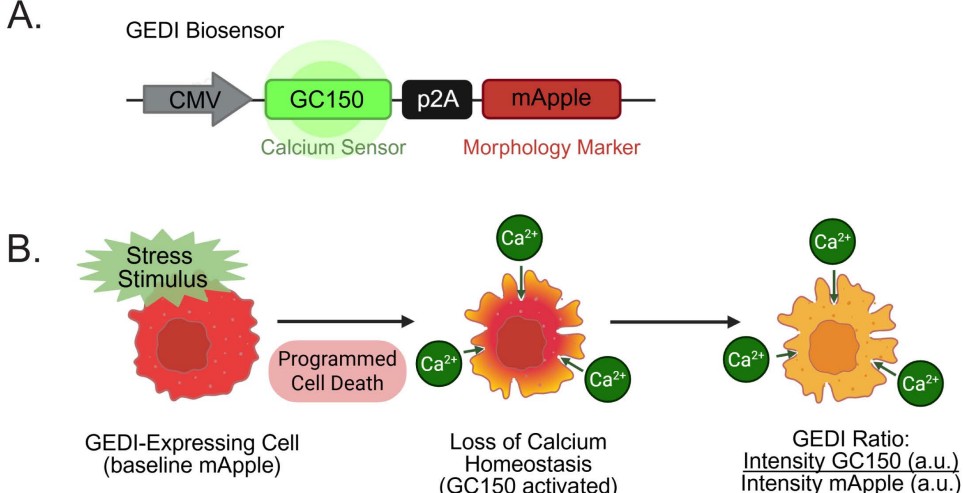

**Fig 1. Schematic of the Genetically Encoded Death Indicator (GEDI) biosensor and detection mechanism. (A)** The GEDI biosensor construct links a calcium sensor (GC150) and a constitutively expressed mApple via a P2A peptide under a CMV promoter. **(B)** The mechanism by which the GEDI biosensor records cell death. A baseline mApple signal provides a morphological reference of a GEDI-expressing cell. Upon a stress stimulus, programmed cell death leads to loss of membrane integrity and increased calcium influx, activating GC150 fluorescence. The GEDI ratio, defined as GC150 fluorescence intensity/mApple fluorescence intensity in arbitrary units (a.u.), provides a quantitative single-cell measure of calcium dysregulation and cell death.

## Cell culture

H3K27M-pDMG cell line SF8628 (SCC127, Millipore Sigma) was cultured in DMEM-High Glucose (D6546, Sigma) with 10% FBS (S11150H, R&D Systems). Cells were routinely tested for mycoplasma with LookOut Mycoplasma PCR Detection Kit (MP0035, Sigma) and used within 10 passages. For sub culturing, cells were lifted with Trypsin-EDTA (25300062, Life Technologies) for 5 minutes at 37°C then quenched with 2x volume of media. Cells were then pelleted at 300 x g for 5 minutes, then resuspended in cell culture media. All cells were maintained at 37°C with 5% $CO_2$. Media was replaced every 3–4 days and cells were split when confluency reached ~70–80%.

## Gateway cloning

To generate an SF8628 GEDI reporter line, the pMe:GC150-p2A-mApple construct (gift from the Finkbeiner Lab, Gladstone Institutes) was cloned into a pLenti-CMV-Puro destination vector (17452, Addgene) using Invitrogen Gateway LR Clonase II (11791020, Life Technologies). The plasmid map can be found in **S2 File**. Lentivirus was produced in HEK293T cells (CRL-3216, ATCC) via co-transfection with psPAX2 (12260, Addgene) and pMD2.g (12259, Addgene) using TransIT reagent (2300, Mirus Bio). Viral supernatants were collected at 48–72 h, concentrated with Lenti-X (631231, Takara Bio), and used to transduce SF8628 cells. Positive populations were enriched by FACS based on mApple and GC150 expression.

## X-ray irradiation

Cells in 96-well plates were irradiated using a XRAD 225XL (PXI Precision X-ray) irradiator with the following beam parameters: 13.3mA, SSD = 40, 225kV, 2.2Gy/min. The irradiation plate was set to a fixed rotation speed. Typical exposure time for 8 Gy was ~3 minutes. All 0 Gy plates were sham irradiated. Volume of media in culture plates was kept at a total volume of 100 µL

## JQ1 preparation

JQ1 (S7110, Selleck Chemicals) was dissolved in DMSO (D8418, Millipore) to 50 mM, followed by a dilution to 10 mM in 200 Proof Ethanol (BP2818−4, Fisher BioReagents) to account for solubility, and used at the indicated concentrations.

## Endpoint viability assays

SF8628 cells were plated at a density of 5,000 cells/well in 96-well optical plates (29444, VWR) with 100 µL of media. Cell viability was measured with Caspase-Glo 3/7 (PAG8091, VWR) or Cell Titer-Glo (PAG7572, VWR) according to the manufacturer's protocol. Luminescence was measured using a Synergy LX Multi-Mode Reader (Biotek). Media-only wells were used as blanks for normalization.

## Live-cell imaging parameters

GEDI-expressing SF8628 cells were plated in 96-well optical plates at densities of ≤5000 cells per well, optimized to minimize cell overlap while maintaining sufficient cell numbers for population-level analysis. Time-lapse imaging was performed by repeatedly acquiring the same fields of view at defined intervals. For radiation-treated cells, imaging was conducted at 0, 3, 6, 12, 18, 24, 36, and 72 hours post-irradiation, while drug-treated cells were imaged at 0, 24, and 72 hours post-treatment.

## GEDI image analysis

Time-lapse imaging of GEDI-expressing cells was conducted using a Lionheart FX imager equipped with red and green filter sets (Biotek), capturing 10 x 10 montages of wells in 96-well plates (29444−008, VWR). Optimal parameters were set at 20X magnification with mApple exposure at 150 ms, and GC150 at 300 ms. Images were processed using Gen5 v3.11 software (Biotek) for background correction, deconvolution, and stitching. GEDI ratios (GC150/mApple) were quantified using ImageJ 1.54p scripts, and empirical thresholds for cell death were determined from high-dose radiation or drug treatment. GEDI ratios were plotted at 0, 3, 6, 12, 18, 24, 36 hours post-stress using the tidyverse package in RStudio. A Rmarkdown file can be found in S1 detailing how to prepare GEDI data for visualization from ImageJ-generated data.

## Cell tracking

Automated tracking was performed in ImageJ using TrackMate v7.14[8] with loG detection and LAP Tracker. Tracking parameters were optimized to link individual cells across successive timepoints, and GEDI ratios were calculated along tracks as described in **S1 File**.

## Statistics

Data are presented as mean+/- standard error (SEM) or ± 95% confidence interval where noted. Statistical analyses were performed using Prism 8 (Graphpad software), R (4.3.1), R tidyverse package (2.0.0), and Trackmate v7.14. An annotated ImageJ script and Rstudio markdown file can be found in the io.protocol (**S1**). All experiments had at least 2 biological replicates and three technical replicates. A one-way ANOVA with Tukey's multiple-comparison post-hoc test was used to compare dose effects within timepoints or a single dose across timepoints. A 2-sample t-test was used to compare GEDI ratios between two timepoints.

## Results and discussion

This protocol demonstrates how to use the Genetically Encoded Death Indicator (GEDI) biosensor to measure therapy-induced cell death in cancer cells. We selected SF8628, a H3K27M pediatric diffuse midline glioma (pDMG) cell line due to its documented intratumoral heterogeneity and prior use in radiosensitization and xenograft studies [10–15]. GEDI was

stably transduced into SF8628 cells and FACS enriched for high mApple expression using appropriate positive and negative controls (S3 Fig).

To establish an empirical GEDI threshold for cell death, we first exposed cells to a high radiation dose (25 Gy), selected solely to reliably induce cell death and ensure a strong GEDI signal based on previous studies [16–18]. Cells were imaged longitudinally at defined intervals post-irradiation (Fig 2A). In these images, cells undergoing cell death displayed bright GC150 activation (yellow), while others remained GC150-negative (red only). To validate that the visual fluorescence shifts corresponded to quantitative shifts in the GEDI ratio, we tracked individual cells from the same field of view across the time course and quantified their GC150/mApple intensities using ImageJ (Fig 2B). Radiosensitive cells showed a rapid rise in GEDI ratio as they underwent cell death, whereas surviving cells maintained stable values. For population-level analysis, mApple-based object masks were used to quantify the fluorescence intensity of the mApple and GC150 channels for all cells in each field of view, using the ImageJ script provided in the protocol (S1 File). At 25 Gy, GEDI ratios were significantly higher at 24 hours post-irradition (hpi) compared to baseline (Fig 2C). Finally, a previously published method was applied to calculate a GEDI death threshold by comparing mean GEDI ratios of cells at 0 hpi (live) and 24 hpi (dead), yielding a threshold of 1.02 (Fig 2D) [9]. This empirically defined threshold separated live from dead cells and was used to classify cell death in subsequent radiation-based experiments with SF8628 cells. Together, these results demonstrate that radiation-induced stress can be quantified with the GEDI ratio in H3K27M-pDMG cells and provide a framework for assessing the heterogeneity of radiation response at clinically relevant doses.

Having established the GEDI threshold for radiation-induced cell death, we next compared the sensitivity of GEDI with a commonly used luminescence-based endpoint assay for apoptosis, Caspase 3/7 Glo [19]. SF8628 GEDI-expressing cells were treated with 8 Gy irradiation, and both GEDI time-lapse imaging and caspase 3/7 signal were measured in parallel from 0 to 36 hpi. Cells were classified as dead if their GEDI ratio exceeded the threshold. (Fig 3A). GEDI analysis revealed a clear, time-dependent increase in the number of dying cells, with a strong linear relationship between the number of GEDI-dead cells and hours post-irradiation ($R^2 = 0.944$) (Fig 3B). In contrast, caspase 3/7 luminescence remained low until 12 hpi and only showed a significant increase at 36 hpi compared to earlier timepoints (Fig 3C). Although caspase activity and GEDI readouts were both significantly correlated with radiation-induced cell death, caspase activity showed a greater variability ($R^2 = 0.71$), reflecting its delayed and less temporally resolved detection of death events. These results demonstrate that GEDI enables higher temporal resolution of radiation-induced cell death than endpoint luminescence-based caspase assays and support its use as a complementary tool alongside other sensitive cell death indicators, including Annexin V-based methods.

Chemotherapy remains a standard treatment strategy for many cancers, but heterogeneous tumor cell populations often contain resistant subclones that survive therapy and drive relapse [20]. To establish a drug-induced GEDI threshold in parallel with our radiation model, we used JQ1, an inhibitor of the bromodomain and extraterminal domain (BET) family of proteins [21]. BET inhibition has shown efficacy in pre-clinical models of H3K27M-pDMG, yet recent studies of the subclonal diversity in these tumors highlight the need to distinguish drug-sensitive from drug-resistant cells at the single-cell level [12,22]. As an initial step, we performed two conventional endpoint assays to identify an appropriate lethal dose for threshold calibration. Caspase 3/7 Glo showed a significant increase in cell death and CellTiter-Glo revealed a significant decrease in cell viability at 50 µM JQ1 (S4 Fig). We then treated SF8628 GEDI expressing cells with 50 µM JQ1 and observed a significant increase in the GEDI ratios between 12–36 hours post-treatment (hpt) compared to baseline (Fig 4A). The highest mean GEDI ratio from this dataset was used to define the JQ1-induced death threshold (1.06). This threshold was slightly higher than the radiation-induced threshold (1.02), and peaked later (36 h vs 18 h), highlighting the need for treatment-specific calibration. We next evaluated lower, screening-relevant doses of JQ1 (1 µM and 10 µM). GEDI detected dose-dependent increases in death as early as 24 hpt (Fig 4B), and linear regression analysis confirmed strong temporal correlations at both doses (1 µM; $R^2 = 0.958$, 10 µM; $R^2 = 0.993$) (Fig 4C). Importantly, resistant cells that failed to cross the GEDI threshold were identified as early as 24 hpt (Fig 4D). These results demonstrate the utility of

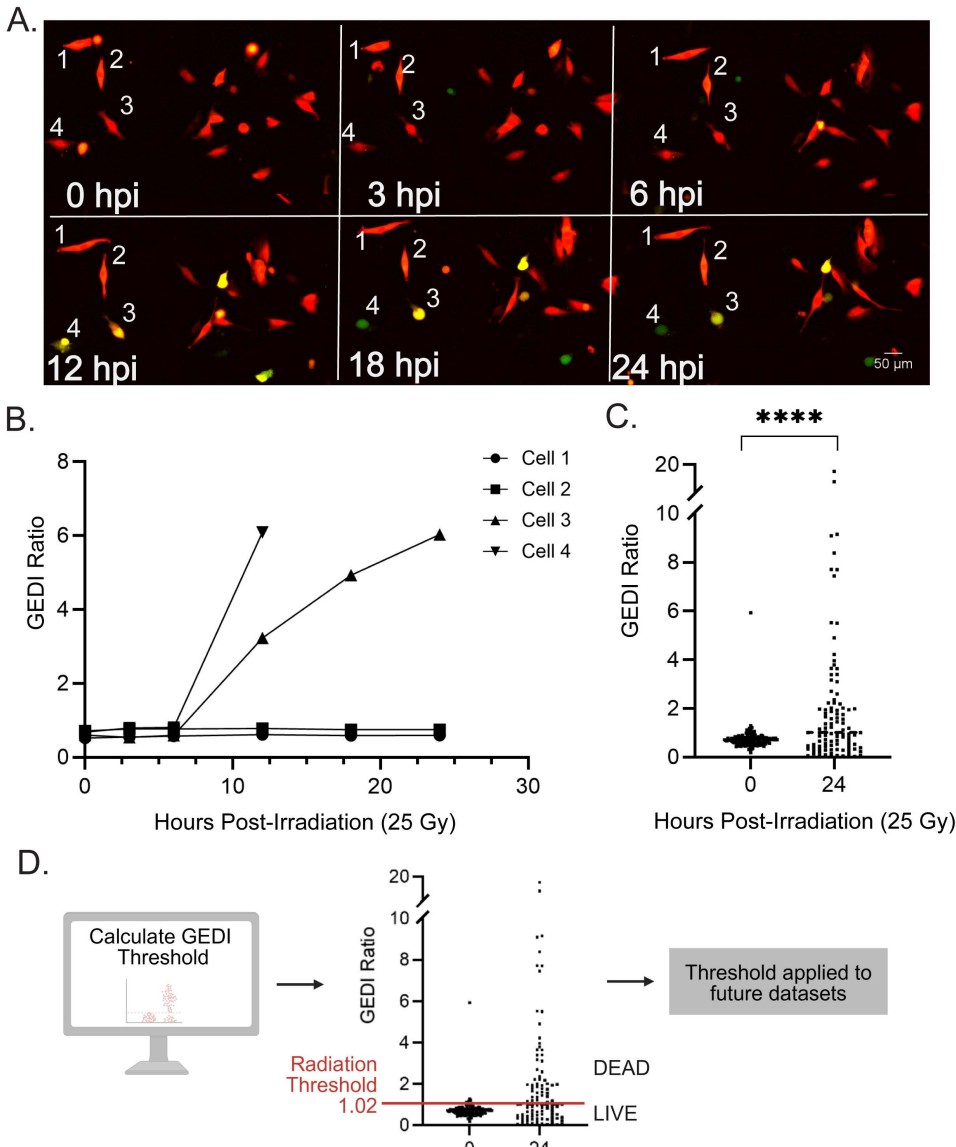

**Fig 2. Determination of a GEDI ratio threshold for radiation-induced cell death. (A)** Representative time-lapse montage of SF8628 GEDI-expressing cells exposed to 25 Gy x-ray irradiation and imaged at 0, 3, 6, 12, 18, and 24 hours post-irradiation (hpi). Numbered cells [1–4] were tracked in B. **(B)** GEDI ratio traces of individual cells from (A) quantified at the indicated timepoints post irradiation. **(C)** GEDI ratio of cells in a 10 x 10, 20X montage field of view at 0 hpi (n = 309 cells) and 24 hpi (n = 266 cells).****p < 0.0001, unpaired t-test. **(D)** The GEDI ratio threshold for cell death was calculated from (C) using *GEDI Threshold = [(Mean GEDI Dead – Mean GEDI Live) x 0.25] + [Mean GEDI Live]* and found to be 1.02. This calculated GEDI threshold is used to determine live and dead cell populations in future experimental conditions. Linear adjustments in ImageJ were made for representative figure images to highlight cellular features of interest. All quantification was done on raw images.

GEDI in detecting drug-sensitive from drug-resistant cells in bulk populations at the single-cell level, enabling rapid and quantitative screening of therapeutic responses.

High-throughput imaging generates large datasets that are difficult to analyze manually, making automated cell tracking essential for extracting dynamic information at the single-cell level. Advances in tracking software have greatly expanded the ability to study heterogeneous cell behaviors over time [23,24]. To test whether GEDI could be incorporated into such

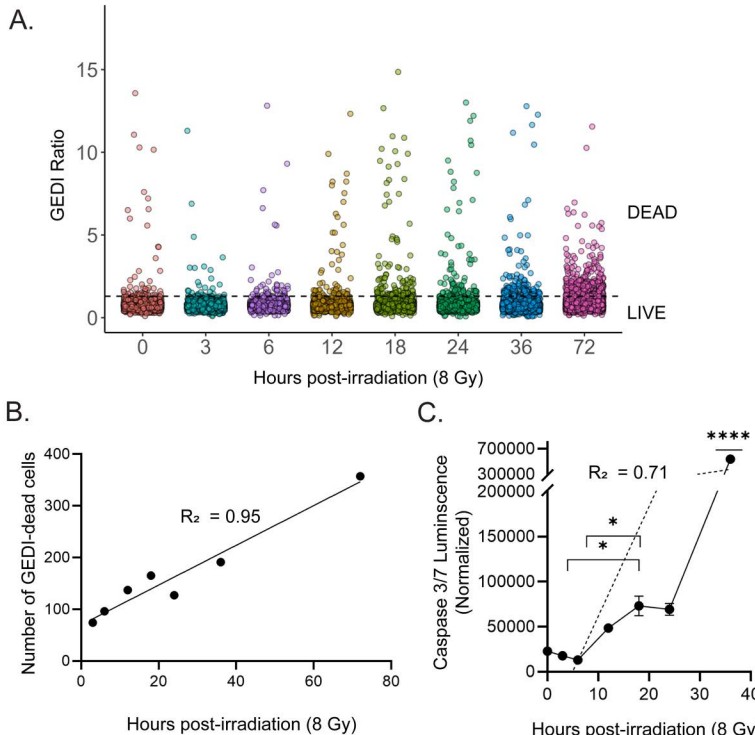

**Fig 3. GEDI captures radiation-induced cell death with higher temporal resolution than endpoint luminescence-based caspase 3/7 detection.**
**(A)** GEDI ratios of SF8628 cells at 0–72 h after a radiation dose of 8 Gy. Each dot represents a single cell (n = 1600–2500 cells per timepoint). The dotted line indicates the radiation-induced GEDI threshold for death determined in Fig 2. **(B)** Linear regression of the number of GEDI-dead cells (above the threshold in **A**). Pearson correlation was used, $R^2 = 0.944$, $p < 0.0001$. **(C)** Caspase 3/7 Glo luminescence at similar timepoints. Signal was normalized to media-only wells. The dashed line represents a linear regression fit ($R^2 = 0.71$). Error bars are SEM (n = 3). Significance was determined by one-way ANOVA followed by Tukey's post-hoc test, comparing all timepoints. Brackets indicate specific pair-wise comparisons that reached significance (*$p < 0.05$), while **** indicates a timepoint significantly different from all others ($p < 0.0001$).

workflows, we applied TrackMate, a user-friendly automated cell tracking plugin available on ImageJ [25]. Integrating GEDI with automated tracking allowed us to generate high-confidence single-cell survival analyses that capture both intrinsic heterogeneity in death timing and potential extrinsic influences from neighboring cells. Our workflow, (Fig 5A; Part IV of the S1 File) consisted of importing hyperstacked images into TrackMate and exporting GEDI ratio data for downstream analysis. Using this pipeline, we quantified single-cell GEDI ratios across radiation doses of 2, 8, and 25 Gy over multiple time points (Fig 5B). GEDI ratios were calculated for each individual cell at each time point, and cells with ratios exceeding the empirically defined threshold were classified as dead, while cells below the threshold were classified as alive. This classification can be used to calculate the absolute numbers and fractions of dead and surviving cells over time. Representative cell tracks from the 8 Gy dataset were also inspected manually to confirm tracking accuracy (Fig 5C). These results demonstrate that GEDI is compatible with open-source automated cell tracking platforms, enabling scalable, unbiased analysis of therapy-induced cell death dynamics.

GEDI's ability to classify live and dying cells extends beyond thresholding into functional and molecular applications (Fig 6). GEDI can be incorporated into high-throughput drug testing to quantify death kinetics, dose-response relationships, and drug or radiation synergies, providing rapid and quantitative readouts of therapeutic efficacy. It also allows for the identification and enrichment of treatment-resistant subclones, a crucial step in understanding tumor heterogeneity and how rare cells survive to drive relapse. These resistant populations can be tracked over time, rechallenged with

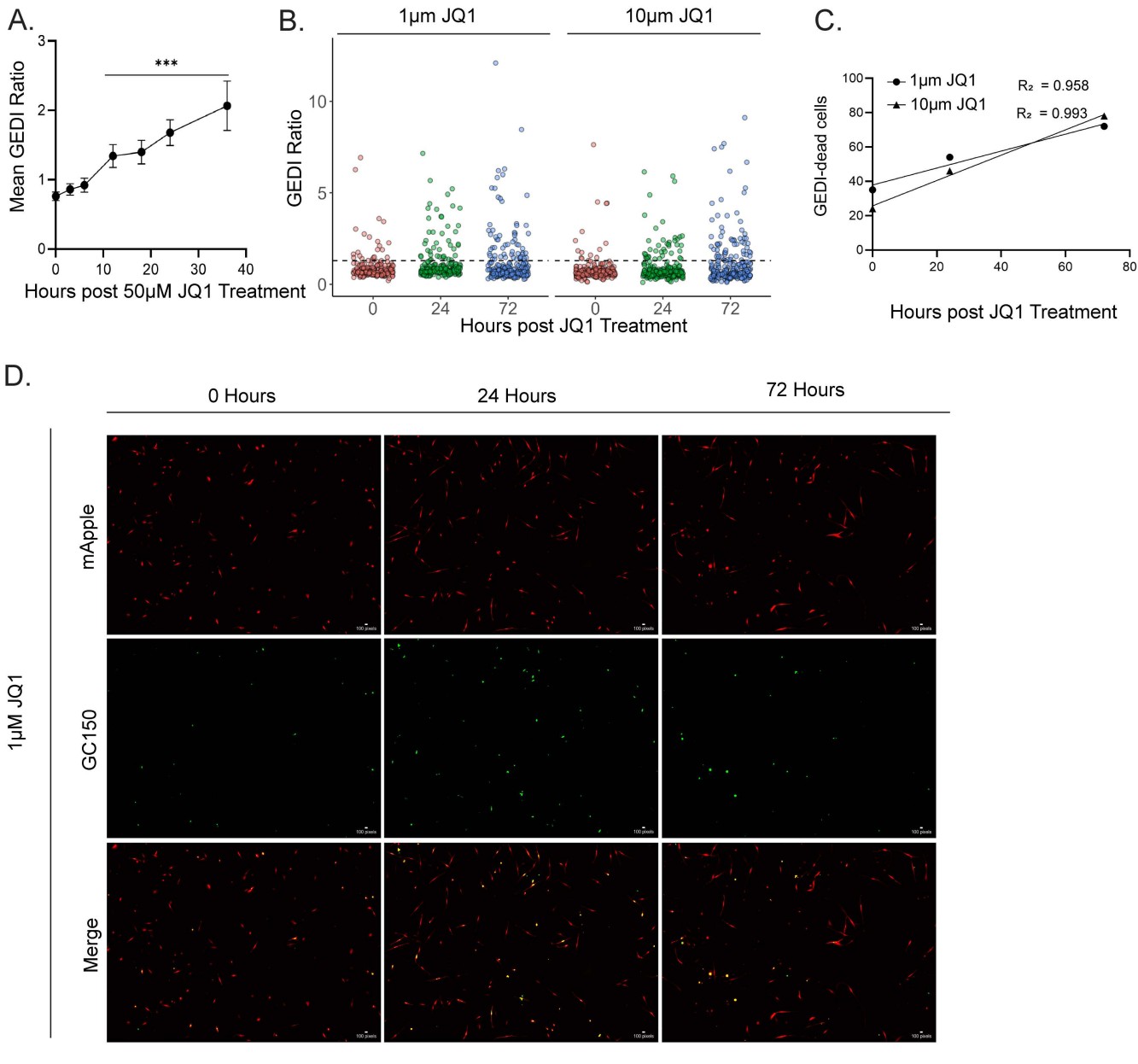

**Fig 4. GEDI detects drug-induced cell death and resistance in SF8628 glioma cells. (A)** Mean GEDI ratios of SF8628 cells treated with 50 μM JQ1 over 0–36 h. The peak ratio (1.06) was used to define the JQ1-induced GEDI threshold. Error bars are 95% confidence intervals, n = 419 cells. Significance by one-way ANOVA followed by a Tukey's post-hoc; each timepoint compared to 0 h. ***p < 0.001 **(B)** Dot plot of GEDI ratios from SF8628 cells treated with 1 μM and 10 μM JQ1 at three timepoints. Dashed line represents GEDI threshold from **(A)**. **(C)** Linear regression of GEDI-dead cells over time for 1 μM ($R^2$ = 0.958) and 10 μM ($R^2$ = 0.993) JQ1 treatments. **(D)** Representative images of SF8628 GEDI cells treated with 1 μM JQ1 at 0, 24, and 72 hours post treatment. Linear adjustments in ImageJ were made for representative figure images to highlight cellular features of interest. All quantification was done on raw images.

secondary treatments, or directly compared to dying cell populations to uncover the mechanisms of persistence. Finally, GEDI-expressing cells can be advanced into downstream biological and molecular assays, including xenograft, explant, or organoid models to test resistance within a tumor microenvironment, or cells can be subjected to transcriptomic,

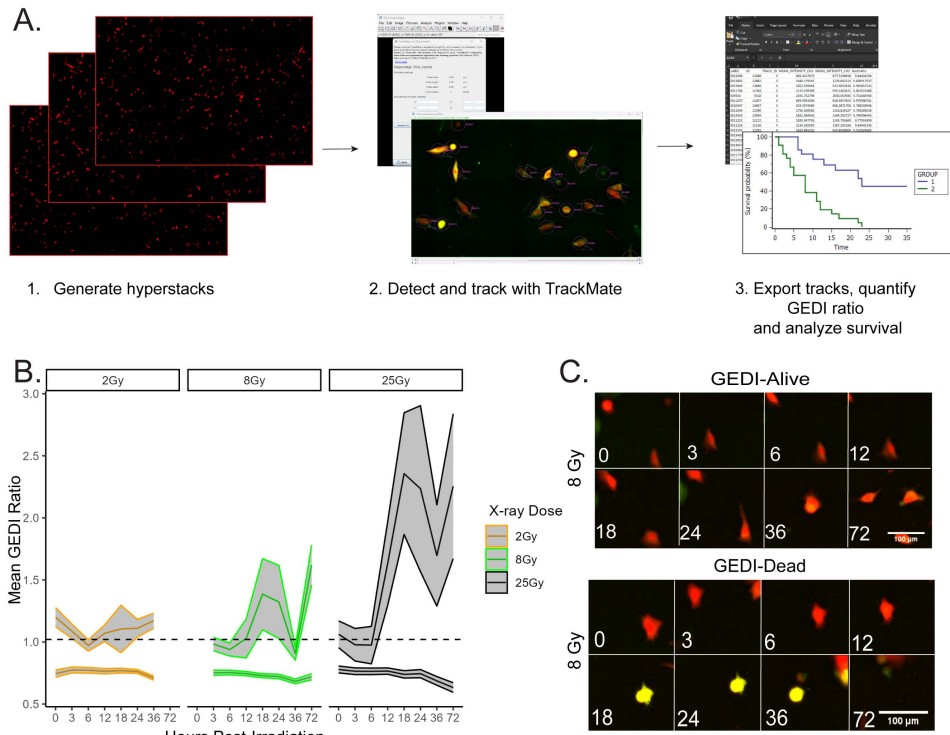

**Fig 5. GEDI is compatible with TrackMate for automated quantification and live cell tracking. (A)** Workflow of single-cell tracking of SF8628 GEDI-expressing cells using TrackMate. Hyperstacks are generated, cells are detected and tracked, and GEDI ratios are quantified to generate survival data. **(B)** Mean GEDI ratios of cells treated with 2, 8, or 25 Gy radiation and tracked using Trackmate for seven time intervals post-irradiation (n = 200-300 cells per condition). Shaded areas represent 95% confidence intervals. The dashed line represents the radiation-induced GEDI threshold (see Fig 2), which is used in downstream analyses to classify individual cells as live or dead and to calculate absolute numbers and fractions of dying cells over time. **(C)** Representative TrackMate-generated montage images of single tracks from the 8 Gy irradiated cells. Shown are examples of a cell that remains below the GEDI threshold ("Alive") and a cell that exceeded the threshold ("Dead"). Linear adjustments in ImageJ were made for representative figure images to highlight cellular features of interest. All quantification was done on raw images.

epigenomic, or gene perturbation studies to define pathways of therapeutic response. By linking dynamic cell death measurements with functional and molecular characterization, GEDI provides a versatile platform to dissect therapeutic response, resistance, and clonal evolution in cancer.

In summary, this study demonstrates that a biosensor originally developed to detect neurodegeneration can be repurposed to screen for radio- and chemo-resistant cancer cells. Using SF8628, a treatment-naive H3K27M-pediatric glioma cell line, we established radiation- and drug-induced thresholds for cell death and showed that GEDI provides both temporal and spatial resolution of death events in heterogeneous populations. We further demonstrate that GEDI is compatible with open-source automated single-cell tracking, enabling high-throughput single-cell survival analyses. Although this protocol is illustrated using SF8628 cells, GEDI has also been applied to an independent H3K27M-pDMG cell line and is expected to be broadly adaptable to additional cancer types following empirical threshold calibration [26].

While GEDI offers a highly sensitive approach for screening radiation- and chemotherapy-resistant cancer cells, it is important to consider its use in the context of other available cell death reporters, as well as several technical limitations. Other live-cell apoptosis reporters, such as caspase or Annexin V fluorescence-based assays, represent complementary approaches for detecting apoptosis, but introduce additional reagent cost, phototoxicity, and are often

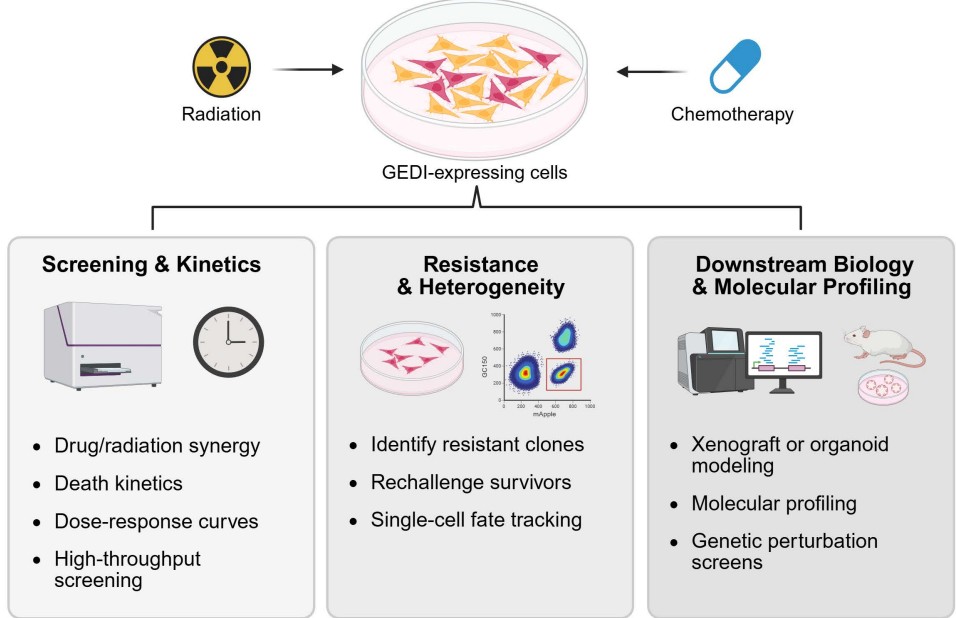

**Fig 6. Applications of GEDI-expressing cells in cancer research.** After treatment with radiation or chemotherapy, GEDI-expressing cells can be routed into downstream workflows to quantify therapeutic efficacy, dissect resistance mechanisms, and inform cancer biology and treatment design at single-cell resolution.

less compatible with long-term or repeated imaging workflows than genetically encoded reporters such as GEDI. In contrast, the performance of GEDI depends primarily on maintaining consistent imaging parameters from threshold calibration through experimental runs to ensure accurate quantification. We have utilized Gen5's image processing tools, however, ImageJ offers several similar algorithms. Careful calibration of exposure settings is required for optimal signal-to-noise; for example, best results were achieved in our system when GC150 exposure was set at two times the exposure of mApple. Imaging systems must allow repeated acquisition of the same field of view to support time-lapse imaging. Empirical determination of a GEDI threshold should also be done at each treatment condition, as shown by the slight differences between radiation- and drug-induced thresholds and the time required to reach a maximum. Additionally, combination treatments may need further calibration. Finally, TrackMate's preset spot and tracking detectors can generate errors, so manual curation of cell tracks or the use of more advanced tracking algorithms are recommended to improve accuracy.

Despite these limitations, GEDI markedly advances the ability to detect therapy-induced cell death at single-cell resolution. Compared with traditional endpoint assays, GEDI captures early and dynamic death events, distinguishes resistant subclones, and provides a flexible platform that can be paired with downstream functional and molecular analyses. This sensitivity has the potential to accelerate preclinical drug and radiation screening and to deepen our understanding of heterogeneity and resistance mechanisms in cancer.

## Supporting information

**S1 File. Step-by-step GEDI live-cell imaging and analysis protocol, also available on protocols.io.** This file contains the detailed protocols.io workflow for generating GEDI-expressing cells, performing live cell imaging, calculating GEDI ratios, and conducting TrackMate-based single-cell tracking and downstream analysis.
(PDF)

**S2 File. Full plasmid sequence of pLenti-CMV-GC150-P2A-mApple expression construct.** This file contains the complete annotated plasmid sequence used to generate the GEDI-expressing cells.
(DOCX)

**S3 Fig. FACS gating strategy for enrichment of GEDI-expressing SF8628 cells.** (**A**) Side scatter area (SSC-A) versus forward scatter area (FSC-A) to exclude debris. (**B**) Forward scatter height (FSC-H) versus FSC-A gating to select single cells. (**C**) Gating strategy to exclude GC150-high cells using heat-shocked cells as a positive control, defining a viable GC150-low population for sorting (**D**) From the GC150-low population, gating strategy to enrich for mApple-positive cells used for downstream experiments.
(TIF)

**S4 Fig. Endpoint apoptosis and viability assays used to inform GEDI threshold calibration (A) Normalized Caspase 3/7 Glo luminescence from SF8628 cells treated with increasing JQ1 concentrations and measured 24 hours post-treatment.** Error bars represent SEM, $n = 3$. Significance was determined by a one-way ANOVA followed by a Tukey's post hoc, ** $p < 0.01$. (**B**) Normalized CellTiter-Glo luminescence from SF8628 cells treated at increasing JQ1 concentrations and measured 24 hours post-treatment. Error bars represent SEM, $n = 3$. Significance was determined by a one-way ANOVA followed by a Tukey's post hoc,* $p < 0.05$,*** $p < 0.001$.
(TIF)

## Acknowledgments

The authors acknowledge the University of Kentucky the Flow Cytometry and Immune Monitoring core for the use of their facilities. We thank Steven Finkbeiner (The Gladstone Institutes) for the GEDI plasmid.

## Author contributions

**Conceptualization:** Jessica S. Blackburn, Viral D. Oza.

**Data curation:** Viral D. Oza, Colin S. Williams.

**Formal analysis:** Jessica S. Blackburn, Viral D. Oza.

**Funding acquisition:** Jessica S. Blackburn, Viral D. Oza.

**Investigation:** Viral D. Oza.

**Methodology:** Viral D. Oza, Colin S. Williams.

**Project administration:** Jessica S. Blackburn, Viral D. Oza.

**Resources:** Jessica S. Blackburn.

**Software:** Viral D. Oza.

**Supervision:** Jessica S. Blackburn.

**Validation:** Viral D. Oza.

**Visualization:** Jessica S. Blackburn, Viral D. Oza.

**Writing – original draft:** Jessica S. Blackburn, Viral D. Oza, Colin S. Williams.

**Writing – review & editing:** Jessica S. Blackburn, Viral D. Oza, Colin S. Williams.

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
