## [Decision Letter · Decision Letter 0]

8 Dec 2025

Dear Dr. Blackburn,

Thank you for submitting your manuscript to PLOS ONE. After careful consideration, we feel that it has merit but does not fully meet PLOS ONE’s publication criteria as it currently stands. Therefore, we invite you to submit a revised version of the manuscript that addresses the points raised during the review process.

We look forward to receiving your revised manuscript.

Kind regards,

Shuo Qie

Academic Editor

PLOS One

Journal Requirements:

“National Institutes of Health, Grant/Award Numbers R37CA227656 (to J.S.B.) and F99CA294265 (to V.D.O.) and the Kentucky Pediatric Cancer Research Trust Fund PON27282400002665. This research is also supported by the Flow Cytometry and Immune Monitoring Shared Resource of the Markey Cancer Center (P30CA177558).”

6. Please include captions for your Supporting Information files at the end of your manuscript, and update any in-text citations to match accordingly. Please see our Supporting Information guidelines for more information: http://journals.plos.org/plosone/s/supporting-information .

7. We note you have not yet provided a protocols.io PDF version of your protocol and/or a protocols.io DOI. When you submit your revision, please provide a PDF version of your protocol as generated by protocols.io (the file will have the protocols.io logo in the upper right corner of the first page) as a Supporting Information file. The filename should be S1_file.pdf, and you should enter “S1 File” into the Description field. Any additional protocols should be numbered S2, S3, and so on. Please also follow the instructions for Supporting Information captions [https://journals.plos.org/plosone/s/supporting-information#loc-captions]. The title in the caption should read: “Step-by-step protocol, also available on protocols.io.”

Please assign your protocol a protocols.io DOI, if you have not already done so, and include the following line in the Materials and Methods section of your manuscript: “The protocol described in this peer-reviewed article is published on protocols.io (https://dx.doi.org/10.17504/protocols.io.[...]) and is included for printing purposes as S1 File.” You should also supply the DOI in the Protocols.io DOI field of the submission form when you submit your revision.

If you have not yet uploaded your protocol to protocols.io, you are invited to use the platform’s protocol entry service [https://www.protocols.io/we-enter-protocols] for doing so, at no charge. Through this service, the team at protocols.io will enter your protocol for you and format it in a way that takes advantage of the platform’s features. When submitting your protocol to the protocol entry service please include the customer code PLOS2022 in the Note field and indicate that your protocol is associated with a PLOS ONE Lab Protocol Submission. You should also include the title and manuscript number of your PLOS ONE submission.

Reviewers' comments:

Reviewer's Responses to Questions

**Comments to the Author**



Reviewer #1: Yes

2. Has the protocol been described in sufficient detail?

To answer this question, please click the link to protocols.io in the Materials and Methods section of the manuscript (if a link has been provided) or consult the step-by-step protocol in the Supporting Information files.

Reviewer #1: Yes

3. Does the protocol describe a validated method?

Reviewer #1: Yes

4. If the manuscript contains new data, have the authors made this data fully available?

Reviewer #1: Yes

**5. Is the article presented in an intelligible fashion and written in standard English?**

Reviewer #1: Yes

Reviewer #1: The Genetically Encoded Death Indicator (GEDI) is a ratiometric, dual-fluorescence biosensor that enables real-time detection of cell death through calcium influx. It was originally developed to study neurodegeneration (PMC8421388). In this manuscript, the authors modified GEDI to assess whether it can be used to assess cell death in cancer cells in vitro in cell culture. In order to show GEDI efficacy, the authors expressed the GEDI dual reporter in an H3K27M-pDMG (diffuse midline glioma) cell line and tested whether the GEDI reporter can accurately distinguish dying cells treated with radiation and JQ1, a bromodomain inhibitor that has been shown previously to potentiate killing in combination with radiation (PMC11213469). Radiation therapy is used to treat this disease. Overall, this protocol represents an innovation over existing approaches to assess cell death in live cells via microscopy.

The strengths of this manuscript are:

1. This protocol provides a way to assess the effect of treatment on apoptosis in live cancer cells using live imaging.

2. The data generated from this protocol can be analyzed and quantified using ImageJ software**,** which is free and easily accessible.

3. Based on the data provided, it appears that the GEDI reporter system may be more sensitive than the traditional caspase and CellTiter-Glo protocols.

4. Since the morphology of the cells is captured, if there are changes in the morphology of the resistant and sensitive cells, this protocol can assess these differences. Second, since the cells express a fluorescent protein, these cells could, in principle, be tracked over time in vitro and in vivo.

The limitations of this protocol stem from the following, which will need to be addressed by the authors.

Major considerations:

1. Use of a single cancer cell line in the analysis. Since this is a new cell death reporter that has not been used before in cancer cells, comparing among different cancer cell lines would greatly increase the utility and clarify the limitations for using this approach. Can the authors select another cell line from the same tumor type or another solid tumor to show that the data generated are reproducible?

2. The comparisons of the GEDI reporter to the CellTiter-Glo and caspase 3/7-based staining in Figures 3Band C that are routinely used to assess cell death are not "apples to apples" comparisons. There are live cell dyes to detect caspase 3/7 and annexin V staining in live cells using microscopy-imaging-based protocols. This would be a true comparison needed to assess whether the GEDI system is more sensitive than caspase fluorescence detection in cells. This analysis would be needed to show that the GEDI system is equivalent to or better than equivalent assays used in laboratories. Even if the GEDI system is equivalent, it could be advantageous for screening as the caspase and annexin V staining dyes, which are used in conjunction with nuclear-red or other color stains, tend to be expensive, limiting their use in screening.

3. The use of 25 and 8 Gy in the experiments is a problem as patients generally are not given single 25 or 8 Gy radiation doses but rather 25 or 8 Gy are given as cumulative 2-4 Gy daily doses of radiation. Hence, the non-physiological doses of radiation used in the initial characterization may not be representative of the conditions that typical screens may occur under. 4 Gy or cumulative dosing of 4 or 2 Gy over time might be more relevant.

4. Can the authors provide a more in-depth description in the materials and methods section of how the imaging of the cells was conducted? I.e., how often were the cells imaged, how many cells were plated, is the time point gap between serial images sufficient to track cells?

Minor comments:

1. Can the authors provide a figure in Figure 5 that can correlate GEDI ratios to absolute numbers of cells dying over the number of live cells?

2. In Figure 6, the claim that this protocol can be used to perform lineage tracing is not justified by the experiments shown. For lineage tracing, some kind of barcoding system would be needed as all the cells are labelled with the same color and there is no way to distinguish between cells.

**Do you want your identity to be public for this peer review?** For information about this choice, including consent withdrawal, please see our Privacy Policy

Reviewer #1: No

---

## [Author Response · Author response to Decision Letter 1]

13 Jan 2026

RESPONSE TO REVIEWERS

We thank the editor and reviewer for their constructive feedback on our manuscript. We are pleased that the reviewer found the protocol to be clearly described, validated, and of utility to the research community. Below, we address each comment point-by-point and describe the corresponding revisions made to the manuscript.

Editor Comments:

E-Q1: Please ensure that your manuscript meets PLOS ONE's style requirements, including those for file naming.

Author: The manuscript has been reformatted to fully comply with the PLOS ONE style and file-naming requirements, using the journal-provided templates.

E-Q2: Please note that funding information should not appear in any section or other areas of your manuscript. We will only publish funding information present in the Funding Statement section of the online submission form. Please remove any funding-related text from the manuscript.

Author: All funding-related text has been removed from the manuscript.

E-Q3. We note that the grant information you provided in the ‘Funding Information’ and ‘Financial Disclosure’ sections do not match. When you resubmit, please ensure that you provide the correct grant numbers for the awards you received for your study in the ‘Funding Information’ section.

Author: Funding information has been reviewed and corrected.

E-Q4. Thank you for stating the following financial disclosure: “National Institutes of Health, Grant/Award Numbers R37CA227656 (to J.S.B.) and F99CA294265 (to V.D.O.) and the Kentucky Pediatric Cancer Research Trust Fund PON27282400002665. This research is also supported by the Flow Cytometry and Immune Monitoring Shared Resource of the Markey Cancer Center (P30CA177558).”

Author: The funders had no role in study design, data collection and analysis, decision to publish, or preparation of the manuscript. This has been updated on the submission form and noted in the cover letter as requested.

E-Q5: When completing the data availability statement of the submission form, you indicated that you will make your data available on acceptance. We strongly recommend all authors decide on a data sharing plan before acceptance, as the process can be lengthy and hold up publication timelines. Please note that, though access restrictions are acceptable now, your entire data will need to be made freely accessible if your manuscript is accepted for publication. This policy applies to all data except where public deposition would breach compliance with the protocol approved by your research ethics board. If you are unable to adhere to our open data policy, please kindly revise your statement to explain your reasoning and we will seek the editor's input on an exemption. Please be assured that, once you have provided your new statement, the assessment of your exemption will not hold up the peer review process.

Author: This has been updated on the submission form to state: All data underlying the findings are available within the manuscript, Supporting Information files, and the publicly accessible GitHub repository ‘GEDI_Cancer’, which contains representative raw images, processed datasets, and all analysis scripts. Full raw imaging datasets are available from the corresponding author without restriction.

E-Q6: Please include captions for your Supporting Information files at the end of your manuscript, and update any in-text citations to match accordingly. Please see our Supporting Information guidelines for more information: http://journals.plos.org/plosone/s/supporting-information.

Author: Supporting information captions have been added to the end of the manuscript in accordance with PLOS ONE guidelines.

E-Q7: We note you have not yet provided a protocols.io PDF version of your protocol and/or a protocols.io DOI. When you submit your revision, please provide a PDF version of your protocol as generated by protocols.io (the file will have the protocols.io logo in the upper right corner of the first page) as a Supporting Information file. The filename should be S1_file.pdf, and you should enter “S1 File” into the Description field. Any additional protocols should be numbered S2, S3, and so on. Please also follow the instructions for Supporting Information captions [https://journals.plos.org/plosone/s/supporting-information#loc-captions]. The title in the caption should read: “Step-by-step protocol, also available on protocols.io.”

Please assign your protocol a protocols.io DOI, if you have not already done so, and include the following line in the Materials and Methods section of your manuscript: “The protocol described in this peer-reviewed article is published on protocols.io (https://dx.doi.org/10.17504/protocols.io.[...]) and is included for printing purposes as S1 File.” You should also supply the DOI in the Protocols.io DOI field of the submission form when you submit your revision.

If you have not yet uploaded your protocol to protocols.io, you are invited to use the platform’s protocol entry service [https://www.protocols.io/we-enter-protocols] for doing so, at no charge. Through this service, the team at protocols.io will enter your protocol for you and format it in a way that takes advantage of the platform’s features. When submitting your protocol to the protocol entry service please include the customer code PLOS2022 in the Note field and indicate that your protocol is associated with a PLOS ONE Lab Protocol Submission. You should also include the title and manuscript number of your PLOS ONE submission.

Author: We have updated the text in the manuscript providing the protocols.io link and submitted the PDF version with our revision.

Reviewer 1

R1-Q1: Use of a single cancer cell line in the analysis. Since this is a new cell death reporter that has not been used before in cancer cells, comparing among different cancer cell lines would greatly increase the utility and clarify the limitations for using this approach. Can the authors select another cell line from the same tumor type or another solid tumor to show that the data generated are reproducible?

Author: We agree that demonstrating applicability across multiple cancer cell lines would further broaden the utility of the GEDI biosensor. Because this manuscript is intended as a protocol-focused study, we prioritized rigorous methodological validation and workflow development in a single, well-characterized H3K27M-pDMG model rather than breadth across tumor types. Importantly, we have independently applied GEDI to a second pDMG line (SF7761) to quantify radiation-induced single-cell death dynamics in our recently published study (PMID: 41167985), which we now cite in the manuscript (lines 406-409).

R1-Q2: The comparisons of the GEDI reporter to the CellTiter-Glo and caspase 3/7-based staining in Figures 3B and C that are routinely used to assess cell death are not "apples to apples" comparisons. There are live cell dyes to detect caspase 3/7 and annexin V staining in live cells using microscopy-imaging-based protocols. This would be a true comparison needed to assess whether the GEDI system is more sensitive than caspase fluorescence detection in cells. This analysis would be needed to show that the GEDI system is equivalent to or better than equivalent assays used in laboratories. Even if the GEDI system is equivalent, it could be advantageous for screening as the caspase and annexin V staining dyes, which are used in conjunction with nuclear-red or other color stains, tend to be expensive, limiting their use in screening.

Author: We agree that live-cell caspase or Annexin V fluorescence assays would provide a more direct single-cell comparison to GEDI for microscopy-based detection of apoptosis. Our intent in Figures 3B-C was not to benchmark GEDI against all available cell-death reporters, but rather to compare it to a widely used endpoint luminescence-based assay that is commonly employed in high-throughput screening workflows. We have revised the manuscript text and figure legend to clarify this distinction and to avoid overinterpretation of relative sensitivity. We now emphasize that GEDI provides improved temporal resolution compared to endpoint luminescence assays (lines 198-202) , while acknowledging that live-cell fluorescence-based apoptosis reporters represent an alternative class of tools with different experimental trade-offs (lines 307-310).

R1-Q3: The use of 25 and 8 Gy in the experiments is a problem as patients generally are not given single 25 or 8 Gy radiation doses but rather 25 or 8 Gy are given as cumulative 2-4 Gy daily doses of radiation. Hence, the non-physiological doses of radiation used in the initial characterization may not be representative of the conditions that typical screens may occur under. 4 Gy or cumulative dosing of 4 or 2 Gy over time might be more relevant.

Author: We agree that clinical radiotherapy is delivered as fractionated dosing rather than single high-dose exposures. In this protocol, 25 Gy was used solely as a calibration condition to empirically define a GEDI death threshold and was not intended to represent a clinically relevant dosing scheme. We have clarified this point in the manuscript (line 160-161). Importantly, we demonstrate GEDI performance across a range of radiation doses, including a clinically relevant per-fraction dose (2 Gy, Fig 5). Because GEDI thresholding is empirically defined, investigators applying GEDI to fractionated or cumulative dosing regimens should calibrate thresholds for specific dose conditions used, which we noted in lines 316-318. Accordingly, the protocol is compatible with clinically relevant fractionation schemes without being limited to the single-dose examples shown here.

R1-Q4: Can the authors provide a more in-depth description in the materials and methods section of how the imaging of the cells was conducted? I.e., how often were the cells imaged, how many cells were plated, is the time point gap between serial images sufficient to track cells?

Author: We have expanded the materials and methods section to explicitly describe live-cell imaging parameters and automated tracking conditions, including imaging frequency, plating density, and tracking methodology (lines 124-128; line 142). The time point gap between images is sufficient to track cells.

R1-Q5: Can the authors provide a figure in Figure 5 that can correlate GEDI ratios to absolute numbers of cells dying over the number of live cells?

Author: GEDI ratios are calculated on a per-cell basis in Trackmate and compared to an empirically defined threshold to classify cells as live or dead. This classification can then be used to calculate absolute numbers and fractions of dead and surviving cells over time. We have clarified this workflow in the Results text (lines 255-259) and revised the Fig 5 legend (lines 269-271) to more clearly explain how GEDI ratios are used in downstream analyses to derive cell counts and survival metrics.

R1-Q2: In Figure 6, the claim that this protocol can be used to perform lineage tracing is not justified by the experiments shown. For lineage tracing, some kind of barcoding system would be needed as all the cells are labelled with the same color and there is no way to distinguish between cells.

Author: We would like to thank the reviewer for correctly pointing out that lineage tracing requires prior cell barcoding. We have revised Fig 6 to remove the term “lineage tracing” and replaced with “fate tracking” to more accurately reflect GEDI capabilities.

---

## [Decision Letter · Decision Letter 1]

1 Feb 2026

A live cell biosensor protocol for high-resolution screening of therapy-resistant cancer cells

PONE-D-25-50553R1

Dear Dr. Jessica S. Blackburn,

We’re pleased to inform you that your manuscript has been judged scientifically suitable for publication and will be formally accepted for publication once it meets all outstanding technical requirements.

Kind regards,

Shuo Qie

Academic Editor

PLOS One
---

## [Editor Report · Acceptance letter]

PONE-D-25-50553R1

PLOS One

Dear Dr. Blackburn,

I'm pleased to inform you that your manuscript has been deemed suitable for publication in PLOS One. Congratulations! Your manuscript is now being handed over to our production team.

Kind regards,

on behalf of

Dr. Shuo Qie

Academic Editor

PLOS One